# Deep-Learning-Based Drive-by Damage Detection System for Railway Bridges

Donya Hajializadeh 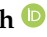

Department of Civil and Environmental Engineering, University of Surrey, Guildford GU2 7XH, UK; d.hajializadeh@surrey.ac.uk

**Abstract:** With the ever-increasing number of well-aged bridges carrying traffic loads beyond their intended design capacity, there is an urgency to find reliable and efficient means of monitoring structural safety and integrity. Among different attempts, vibration-based indirect damage identification systems have shown great promise in providing real-time information on the state of bridge damage. The fundamental principle in an indirect vibration-based damage identification system is to extract bridge damage signatures from on-board measurements, which also embody vibration signatures from the vehicle and road/rail profile and can be contaminated due to varying environmental and operational conditions. This study presents a numerical feasibility study of a novel data-driven damage detection system using train-borne signals while passing over a bridge with the speed of traffic. For this purpose, a deep Convolutional Neural Network is optimised, trained and tested to detect damage using a simulated acceleration response on a nominal RC4 power car passing over a 15 m simply supported reinforced concrete railway bridge. A 2D train–track interaction model is used to simulate train-borne acceleration signals. Bayesian Optimisation is used to optimise the architecture of the deep learning algorithm. The damage detection algorithm was tested on 18 damage scenarios (different severity levels and locations) and has shown great accuracy in detecting damage under varying speeds, rail irregularities and noise, hence provides promise in transforming the future of railway bridge damage identification systems.

**Keywords:** SHM; indirect monitoring; damage detection; railway bridge; data-driven

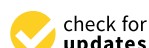

## 1. Introduction

The traditional bridge damage detection approach consists of routine annual visual inspections and more intrusive examinations at six to twelve-year intervals. These inspections are often labour intensive and subjective, as they depend on inspectors' competencies and experience. This approach also results in the degradation of raw data, as there rarely exists a consistent and systematic data collection system. With the growing number of well-aged bridges exceeding their life expectancy and carrying loads beyond their original intended design capacity, bridge owners and operators spend millions of pounds on visual structural health monitoring (SHM) worldwide. To this end, recent years have seen a significant increase in the number of efforts in developing smart SHM systems by directly instrumenting bridges and assessing the structure's condition using direct measurements, such as studies conducted by [1–3].

While these direct SHM systems address some of the shortcomings of visual assessment, their main disadvantages are reliance on prior knowledge of approximate damage location, accessibility for instrumentation and the associated cost of instrumentation and maintenance of the data acquisition system during the monitoring period. These challenges collectively make the application of these systems, for the entire network, logistically difficult and expensive. The direct SHM systems also often require an accurate numerical model of the real structure, which, given the complexity of aged structural behaviour, is a time-consuming process to perfect. Given these limitations, the direct instrumentations are

often bespoke systems, limiting the application of these systems to a specific bridge, which explains the relatively small number of instrumented bridges worldwide.

Collectively, the challenges with visual inspections and direct instrumentations have led to a new set of damage identification techniques entitled 'drive-by' or 'indirect' damage identification systems. The drive-by concept refers to monitoring bridges using measurements from an instrumented vehicle (drive-by vehicle) while passing over the bridge. In other words, the drive-by vehicle acts as an actuator as well as a receiver. The fundamental principle in this approach is that damage-induced physical changes in a structure can manifest in vibration signals measured on a drive-by vehicle. These changes need to be extracted using different signal processing methods and/or vehicle–bridge interaction models to relate the data from the vehicle to the condition of the bridge.

The application of the drive-by concept in bridge damage detection is first introduced by Yang et al. [4], extracting bridge frequencies using acceleration signals measured on a passing vehicle at a speed of 15 kph. The study was then extended to an experimental validation investigation using a one-axle cart, assessing the performance of the drive-by concept as a function of vehicle speed [5]. Later, Yang and Chang [6] used empirical mode decomposition to extract higher frequencies in addition to the fundamental frequency. Oshima et al. [7] expanded this work to investigate the impact of vehicle weight in extracting bridge frequencies.

Yang and Yang [8] and Malekjafarian et al. [9] presented a comprehensive review of damage detection using measurements on a passing vehicle. The study conducted by Malekjafarian et al. [9] notes that the optimal condition for extracting bridge frequencies are low vehicle speed (less than 40 kph), multiple crossing and use of heavy vehicles as actuators. Furthermore, irregularities in road and rail profiles can mask bridge frequencies, as they can excite the vehicle to higher frequencies of the bridge. To remove the blurring effect of the road profile, Yang et al. [10] proposed to use the response from two identical connected vehicles.

As the bridge frequency is a sensitive parameter to operational conditions (varying temperature and vehicle mass), there have been several attempts at utilising other modal parameters for damage detection. For example, McGetrick et al. [11] developed a drive-by damage detection system in which a change of 1% in damping is detectable in acceleration measurement. Yang et al. [12] used Wavelet Transform and Hilbert transform to extract characteristic damage features in the mode shapes. Wavelet Transform approaches have proven to be quite useful in damage indicators. A study conducted by McGetrick and Kim [13] used Continuous Wavelet Transform with Morlet Wavelet to derive a damage indicator. In a similar attempt, Hester and González [14] used Mexican Hat Wavelet to produce a damage detection threshold. In another study, Fitzgerald et al. [15] used Complex Morlet Wavelet for a scour detection indicator by averaging wavelet coefficients between healthy and damaged scenarios.

The majority of the research attempts in drive-by methods have been concentrated on theoretical and experimental model-based damage detection systems under low operational speed (less than 50 kph). The speed of the vehicle is a key parameter in model-based investigations, as speed defines the length of the signal and hence the amount of information stored in the signal. While model-based drive-by approaches have received considerable attention, the application of model-free/data-driven methods has been quite limited. One of the very few data-driven drive-by investigations is the study conducted by Locke et al. [16] building and training a one-dimensional (1D) deep-learning algorithm to develop a damage detection system using the frequency spectrum of simulated acceleration signals on a single-axle quarter-vehicle model, with a maximum vehicle speed of 90 km/h.

Recent years have seen considerable attention towards data-driven damage identification systems using powerful machine learning algorithms. Farrar and Worden et al. have extensively demonstrated the application of data-driven approaches in building damage detection systems for different structures and infrastructures [17–20]. Among different

machine learning algorithms, deep learning approaches have attracted particular interest given their high efficiency and accuracy in object detection and classification.

In general, a typical deep learning algorithm consists of two main components of feature extraction and classification. In feature extraction, a range of signal processing tools is used to extract damage signatures from raw signals. These features are most sensitive to damage state (i.e., detection and classification into severity and location classes). Among different signal processing techniques, Continuous and Discreet Wavelet Transform [21], empirical mode decomposition [22–24], power spectrum and frequency spectrum [16,25] have been widely used as damage-sensitive features. Furthermore, statistical analysis and principal component analysis are often employed in order to reduce and optimise the dimension of the extracted features [26,27].

The second component of a deep learning algorithm involves building and training a classifier algorithm to map selected extracted damage-sensitive features against corresponding damage classes. This task is conducted using a variety of methods, such as multi-perceptron neural networks (MLP) [28–30] and fuzzy inference systems [31]. Since the performance of the algorithm is defined by the efficiency of both these components, it can be deduced that integrating these components in a unit learning body can improve the efficiency of the learning algorithm. This notion has led to a powerful class of deep learning algorithms entitled Convolutional Neural Networks (CNN).

CNN algorithms imitate the functionality of the visual cortex of the brain process in object detection [32]. In this class of deep-learning algorithms, the learning is based on gathering information from neighbouring inputs to form sub-features in the filters as opposed to reshaping multidimensional image data into a 1D feature vector in traditional shallow neural networks [33].

In CNN algorithms, both feature extraction and classification components are built into the architecture of the learning algorithm, reducing the computational efforts in communication between the feature extraction and classification components. In the learning body of a CNN algorithm, the feature extraction component consists of several layers of convolutional and pooling layer pairs. The convolutional layers convert input data, often an image, using filters. In the pooling layers, the in-plane size of feature maps is reduced by down-sampling pixels using a certain strategy to produce deeper representations in successive layers and prevent overfitting [33].

Despite the power of the CNN algorithms, their application in structural damage detection, in particular, in vibration-based approaches, has not been widely reported. For example, the studies conducted by Cha et al. [34], Mohtasham Khani et al. [35], Tong et al. [36] and Kim and Cho [37] used a vision-based technique for crack detection purposes, and Nex et al. [38] reported on the application of vision-based CNN algorithms using remote sensing images. The research studies on vibration-based algorithms have been predominantly focused on 1D CNN algorithms. For example, Sony et al. [39] developed a 1D CNN for a damage localisation system using acceleration signals of the Z24 bridge. Among 2D CNN damage identification systems studies, the 1D time-series responses have either been transformed into two-dimensional (2D) images by resizing the raw data [40] or have used data from multi-sensors to build 2D images [41].

In this study, a 2D deep CNN algorithm is built, trained and tested to detect damage using simulated train-borne signals. A numerical train–track–bridge (TTB) interaction model with an advanced half-car model is built to simulate train-borne accelerations for a range of healthy and damaged scenarios. The simulated accelerations on the front train bogies are then used as initial raw data. The TTB model in this study is used to simulate acceleration time histories only, which can ideally be measured on an instrumented train in practice. It is noteworthy that the TTB model provides no other input to the CNN algorithm.

In summary, the novelty of this study lies in three folds: 1. building a drive-by damage detection system using a 2D CNN algorithm, 2. application of network-in-network CNN architecture for damage detection purposes and 3. using raw real-valued continuous wavelet coefficients as input for a damage detection system. The following sections first

provide an overview of the numerical model used to simulate the train accelerations and the architecture of the CNN algorithm. Then, a brief overview of the Bayesian Optimisation process that was utilised to optimise the architecture of the algorithm is presented. Section 3 presents the application of the approach to several damage scenarios under varying vehicle speeds and discusses the performance of the proposed system, followed by Section 4 with the conclusions drawn.

## 2. Description of the Damage Detection System

### 2.1. TTB Numerical Model

A TTB interaction model couples the dynamic behaviour of three subsystems of train, track and bridge. The number of parameters used in building a TTB model depends on the complexity of the model. In this study, a 2D TTB model, which has been widely used in the literature [4,15,42,43], is used to demonstrate the feasibility of the proposed methodology. A summary of the model and corresponding parameters is presented here. Cantero et al. [44] conducted a comprehensive review of the parameter of this model for all three subsystems.

Figure 1 shows a schematic demonstration of the TTB model and Table 1 summarises the parameters used in this study. As can be seen from Figure 1, the train is represented by half a train carriage (two bogies and a half-car body) simplified by a 10 degrees-of-freedom (10-DOF) system with a combination of lumped masses, rigid bars, springs and dashpots. For the purpose of this study, the parameters for the train model represent a typical RC4 power car.

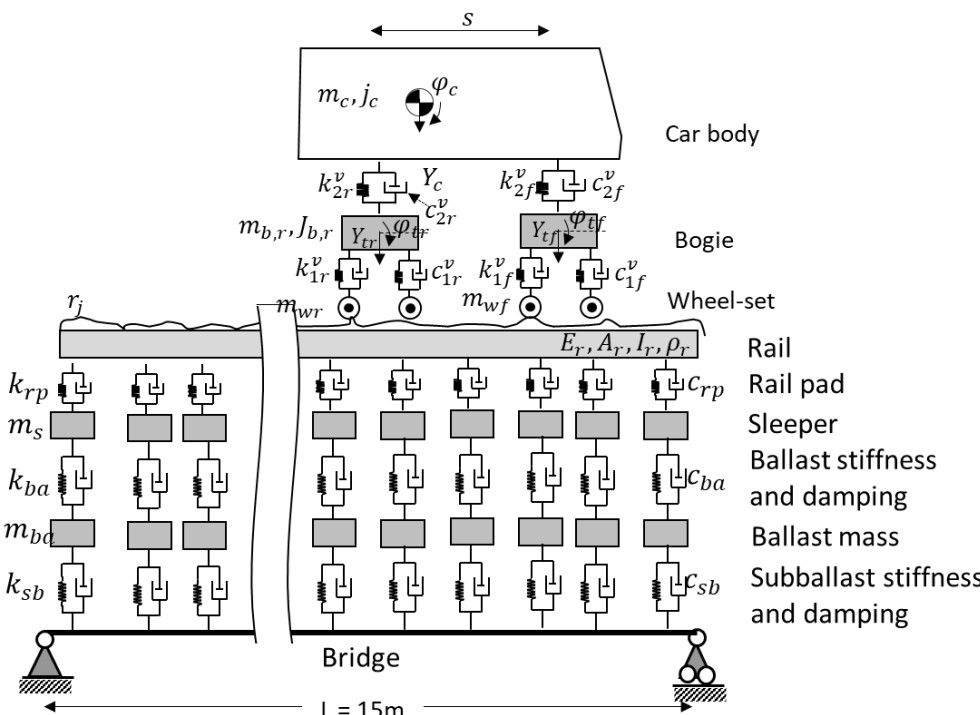

**Figure 1.** Schematic demonstration of the TTB model.

Figure 1 also shows the track and bridge interaction system in which a ballasted railway track is represented by a system of rails, pads, sleepers, ballast and sub-ballast. Similar to the bridge itself, the rail is modelled as discretised Euler–Bernoulli beams resting on a continuously spaced-sprung mass system and the ballast and sleepers are simplified as mass sitting on a system of spring and damping dashpot. In this study, a 15 m simply supported reinforced concrete railway bridge with a density per unit length of 15,000 kg/m$^3$/m and a second moment of area of 0.99 m$^4$ is used to represent the bridge.

To simulate the irregularities of the rail profile, random irregularities are generated using the Federal Railroad Administration (FRA) Power Spectral Density (PSD) function, expressed as Equation (1) [45,46]:

$$S(\omega) = \frac{A_v \omega_2^2 (\omega^2 + \omega_1^2)}{\omega^4 (\omega^2 + \omega_2^2)} \tag{1}$$

where $A_v$ represents the scale factor for the track class. In this study, FRA's class of 4 with $A_v$ of $2.75 \times 10^{-8}$ m$^2$/m$^{-1}$ is used for the irregularities. As for the $\omega_1$ and $\omega_2$ constants, values of $23.294 \times 10^{-3}$ m$^{-1}$ and $13.123 \times 10^{-2}$ m$^{-1}$ are used, respectively, which represent wavelengths in the range of 1.5–305 m.

**Table 1.** Vehicle and track properties.

| Vehicle Properties [47] | | | Track Properties [48] | | |
|---|---|---|---|---|---|
| **Parameter** | **Symbol** | **Value** | **Parameter** | **Symbol** | **Value** |
| Carriage body mass (kg) | $m_c$ | 61,560 | Rail Young's modulus (N/m$^2$) | $E_r$ | $206 \times 10^9$ |
| Carriage body moment of inertia (kg·m$^2$) | $J_c$ | $9.11 \times 10^6$ | Rail cross-sectional area (m$^2$) | $A_r$ | 15.38 |
| Bogie mass (kg) | $m_{br}, m_{bf}$ | 5200 | Rail second moment of area (m$^4$) | $I_r$ | $6.43 \times 10^{-5}$ |
| Bogie moment of inertia (kg·m$^2$) | $J_{br}, J_{bf}$ | 5900 | Rail mass per unit length (kg/m) | $\rho_r$ | 120 |
| Wheelset mass (kg) | $m_{wr}, m_{wf}$ | 1510 | Rail pad stiffness (N/m) | $k_{rp}$ | $80 \times 10^6$ |
| Primary suspension stiffness (N/m) | $k_{1r}^v, k_{1f}^v$ | $4.96 \times 10^6$ | Rail pad damping (N.s/m) | $c_{rp}$ | $60 \times 10^3$ |
| Secondary suspension stiffness (N/m) | $k_{2r}^v, k_{2f}^v$ | $1.9 \times 10^6$ | Mass of sleeper (kg) | $m_s$ | 340 |
| Primary suspension damping (kN·s/m) | $c_{1r}^v, c_{1f}^v$ | 108 | Sleeper spacing (m) | $L_s$ | 0.57 |
| Secondary suspension damping (kN·s/m) | $c_{2r}^v, c_{2f}^v$ | 152 | Ballast stiffness (N/m) | $k_{ba}$ | $120 \times 10^6$ |
| Distance between axles (m) | $L_{ar}, L_{af}$ | 2.7 | Ballast damping (N·s/m) | $c_{ba}$ | $60 \times 10^3$ |
| Horizontal distance between centre of mass of main body and bogie (m) | $L_{cr}, L_{cf}$ | 3.81 | Ballast mass | $m_{ba}$ | 2718 |
| | | | Sub-ballast stiffness (N/m) | $k_{sb}$ | $60 \times 10^6$ |
| | | | Sub-ballast damping (N/m) | $c_{sb}$ | $90 \times 10^3$ |

Assuming that the subscripts *v*, *r* and *b* represent the vehicle, rail and bridge, respectively, and mass, damping and stiffness matrices for each system are denoted by *M*, *C* and *K*, respectively, the coupled system response can be expressed using Equation (2). In this equation, *F* is the external force vector that represents the contribution of gravity and excitation induced by the rail irregularities. The details of deriving the mathematical equations of the coupled system can be found elsewhere [49].

$$
\begin{bmatrix} M_{vv} & 0 & 0 \\ 0 & M_{rr} & 0 \\ 0 & 0 & M_{bb} \end{bmatrix} \begin{Bmatrix} \ddot{Y}_v \\ \ddot{Y}_r \\ \ddot{Y}_b \end{Bmatrix} + \begin{bmatrix} C_{vv} & C_{vr} & 0 \\ C_{rv} & C_{rr} & C_{rb} \\ 0 & C_{br} & C_{bb} \end{bmatrix} \begin{Bmatrix} \dot{Y}_v \\ \dot{Y}_r \\ \dot{Y}_b \end{Bmatrix} + \begin{bmatrix} K_{vv} & K_{vr} & 0 \\ K_{rv} & K_{rr} & K_{rb} \\ 0 & K_{br} & K_{bb} \end{bmatrix} \begin{Bmatrix} Y_v \\ Y_r \\ Y_b \end{Bmatrix} = \begin{Bmatrix} F_v \\ F_r \\ F_b \end{Bmatrix} \tag{2}
$$

In this equation, while the bridge–track coupling terms are constant, the vehicle–track system is time-dependent as it varies with train car position. In the latter system, the DOFs of wheels are merged with the vertical DOFs of the rail; hence, the mass matrix of the track needs to be updated to account for wheel mass at each time step. To solve the coupled equation system, the Newmark-$\beta$ method can be used, as it is shown to be an unconditionally stable numerical approach [4,50]. By solving the coupled system, train-borne acceleration time histories are generated. For this study, accelerations measured at the front bogie are used as raw initial input values. The results are repeatable for the rear bogie accelerations.

To simulate train-borne accelerations for a damaged condition as well as a healthy state, train accelerations are simulated for a range of damage scenarios. In this study, a damage is modelled as a reduction in flexural stiffness of the beam elements with the damage intensity ranging from 5% to 55% at three different locations of quarter-span, mid-span and three-quarter-span. The damage intensity represents the reduction in flexural stiffness of beam elements with an assumed effective damage length of 0.55 m. To demonstrate the intensity of the damage levels on modal properties of the selected beam, the change in the fundamental natural frequency of the bridge is presented in Figure 2. As can be expected, damage at mid-span with a similar level of intensity to the other two locations can result in a much greater change in fundamental natural frequency.

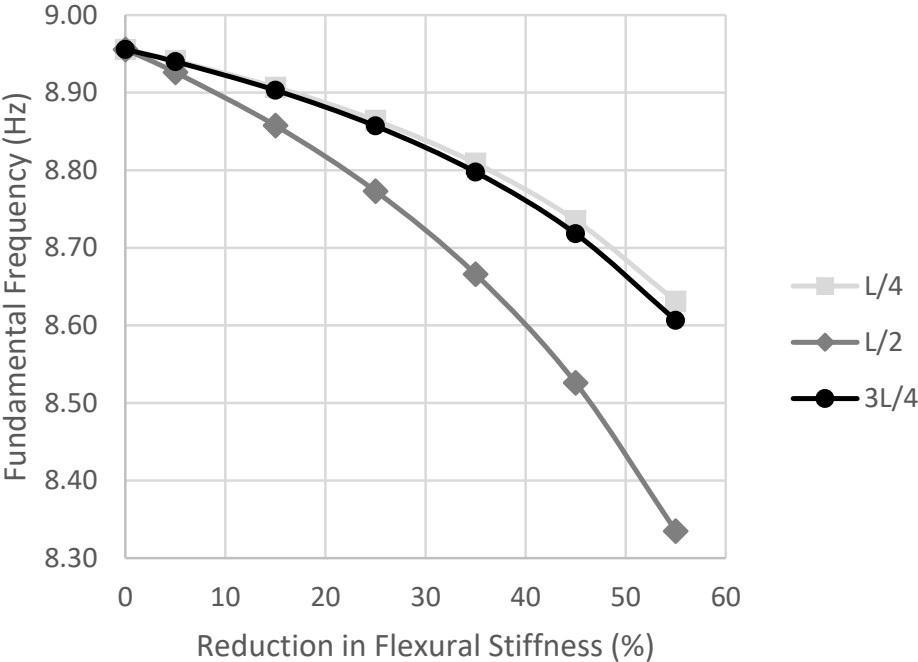

**Figure 2.** Change in fundamental frequency given different damage levels.

To account for speed variability in practice, acceleration time histories were simulated for 100 randomly generated speeds with a mean of 100 kph and covariance of 10%. This resulted in a total number of 21,000 simulated acceleration signals. Figure 3 shows a sample of simulated signals for both bogies under healthy state and different damage scenarios and the vehicle speed of 105 kph. The difference between healthy and damaged signals in the front bogie is 0.06 m/s$^2$ at the maximum damage level of 55% and is 0.0097 m/s$^2$ at 5% damage.

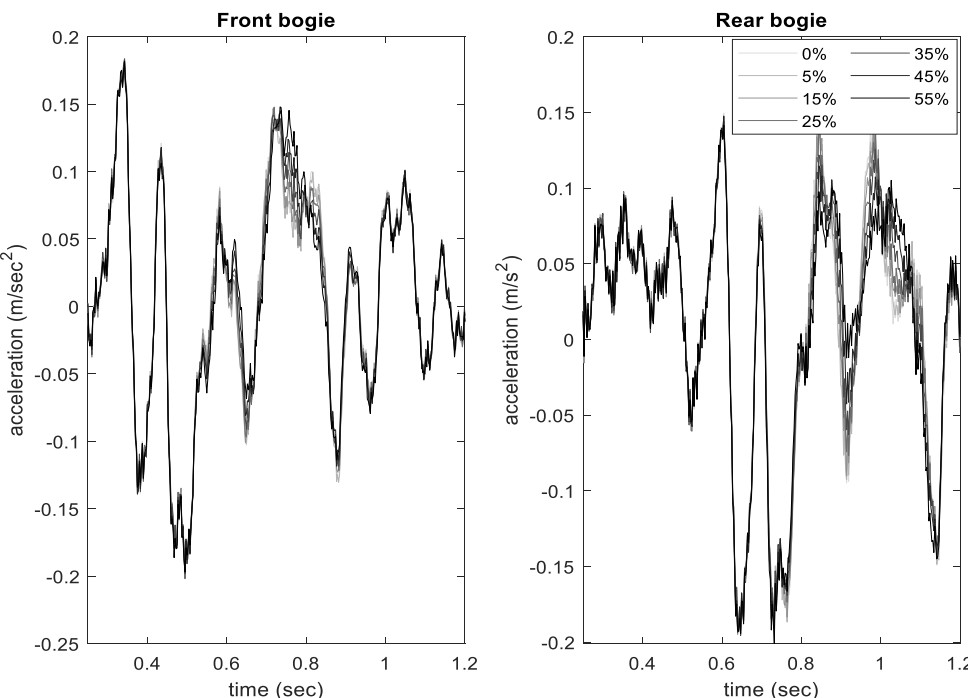

**Figure 3.** A sample of acceleration signals for front (1st) and rear (2nd) bogies for different damage levels at mid-span at the speed of 105 kph.

To provide a stronger damage-sensitive feature with more discriminating power, Continuous Wavelet Transform (CWT) with Morse Wavelet was used. The Fourier transform of the generalised Morse Wavelet can be represented by Equation (3) [51]:

$$\Psi_{P,\gamma}(\omega) = U(\omega) a_{P,\gamma} \omega^{\frac{P^2}{\gamma}} e^{-\omega\gamma} \tag{3}$$

where $U(\omega)$ is the unit step, $a_{P,\gamma}$ is normalising constant, $P^2$ is the time-bandwidth product and $\gamma$ is the symmetry of the Morse Wavelet. For this study, Morse Wavelet with symmetry parameter of 3 and time-bandwidth product of 60 were used. The real-valued Morse Wavelet (real part of the Complex Morse Wavelet) was then used as an input to the CNN algorithm. Real wavelet coefficients are often used for SHM purposes [15,52]. Figure 4 demonstrates an example of real-valued CWT coefficients for a speed of 105 kph for different levels of damage at mid-span and sampling frequency of 400. As can be seen from Figure 4, the difference between healthy and damage state is not visually noticeable. To better highlight the difference, Figure 5 shows the relative difference between real-valued CWT coefficients of healthy and damage scenarios under the same speed for different damage intensities. As can be seen from this figure, the difference between healthy and damage scenarios are more pronounced in a frequency range of 5–10 Hz and more distinct in damage levels of 35–55%. Although Figure 5 can represent a much stronger damage-sensitive input, in practice the relative difference under the same operational condition (e.g., same speed) rarely exists; hence, for training the algorithm, the actual real-valued CWT coefficients are used as input images (examples provided in Figure 4), which showcases the power of the proposed algorithm in accentuating damage-sensitive features that are not visually discernible.

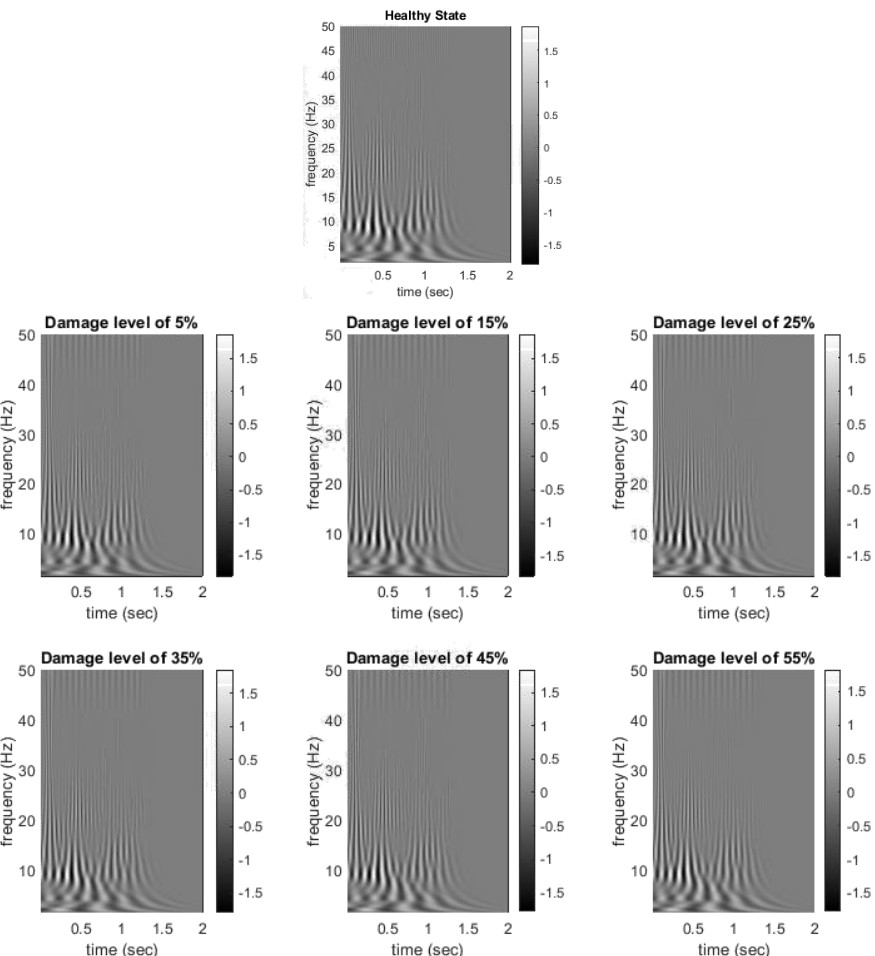

**Figure 4.** A sample of real-valued CWT coefficients for six damage levels and speed of 105 kph.

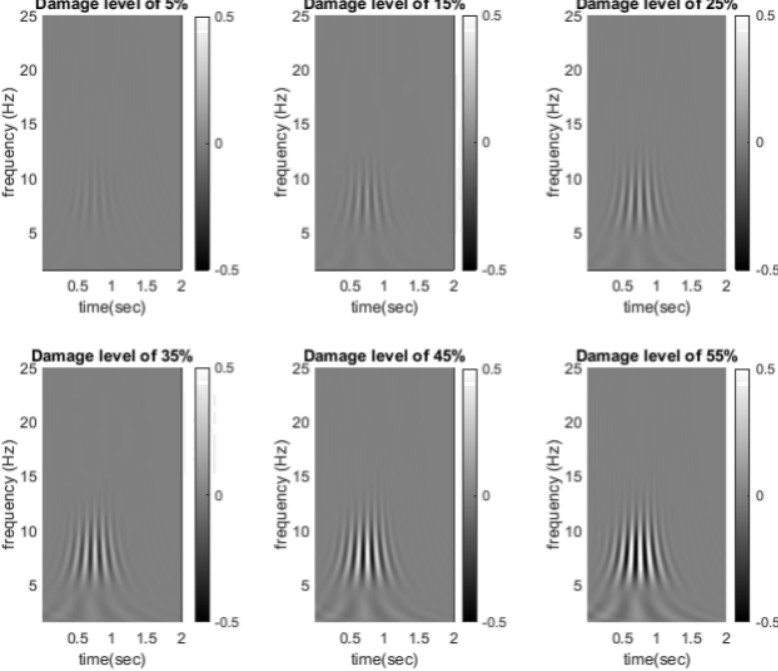

**Figure 5.** Difference between the real-valued CWT coefficients of healthy and six damage levels under speed of 105 kph.

## 2.2. Deep Leaning Architecture

As mentioned in Section 1, a typical CNN architecture consists of layers of convolution, pooling and activation filters followed by fully connected classification layers. Figure 6 shows a schematic demonstration of a typical CNN architecture with dropout layers. The combination of the convolutional and pooling layer pairs forms the feature extraction element of the network. The function of the convolution layer is similar to digital filters by converting an image to a new image which is often referred to as feature maps. These maps aim to accentuate the unique features of the input image. The convolution filters are determined through the training process of the algorithm. On the other hand, the pooling layer combines neighbouring pixels into a single pixel to reduce the dimension of the input image and hence reduce the computational costs. The feature maps are then processed through activation layers which are identical to that of an ordinary multi-perceptron neural network.

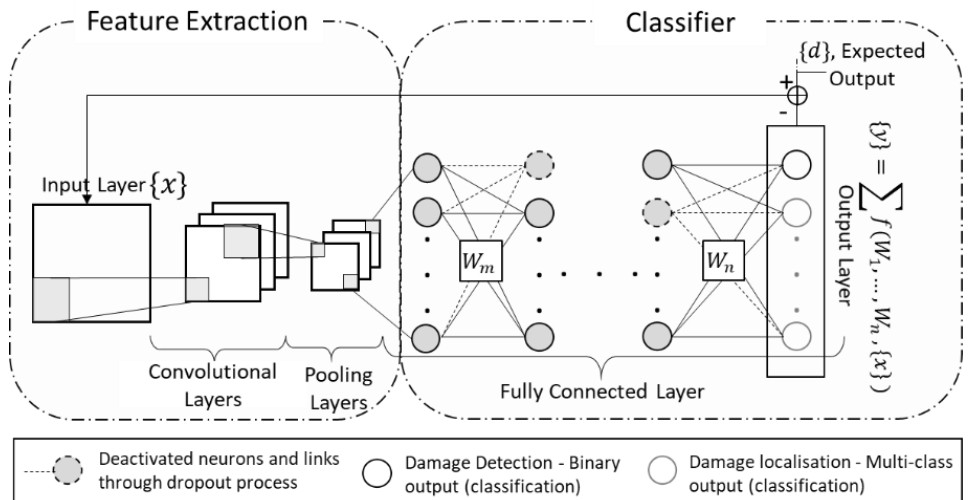

**Figure 6.** Schematic demonstration of a typical CNN architecture.

The classification component of CNN architecture is similar to the architecture of a typical multi-class classification neural network with hidden connected layers, activation layers and often dropout layers. The latter prevents overfitting by randomly zeroing activations or deactivating the nodes and weights during the forward pass in the training process.

In CNN networks, the depth of the structure often defines the performance of the algorithm. Deeper architecture comes with a significant increase in computational costs, which has led to numerous attempts to find a more balanced trade-off between accuracy and computational costs. One of such attempts has led to the development of the GoogLeNet model [53], the winner of the ImageNet Large-Scale Visual Recognition Challenge in 2014. GoogLeNet is a 22-layer deep CNN that consists of 60 convolution layers. The predominant feature of GoogLeNet architecture is its use of the network-in-network approach first proposed by Lin et al. (2013). In this approach, additional $1 \times 1$ convolutional layers are added to the network to increase the depth and the width of the network without a significant drop in performance. In the network architecture of GoogLeNet, the convolution layers are used as dimension reduction modules which aim to remove computational bottlenecks [54]. This has been performed using the introduction of inception modules in the architecture, which contains different sizes of convolutions and pooling filters that provide means of extracting more information into a smaller layer by widening the layer of the neuron network. The structure of an example of an inception module is shown in Figure 7.

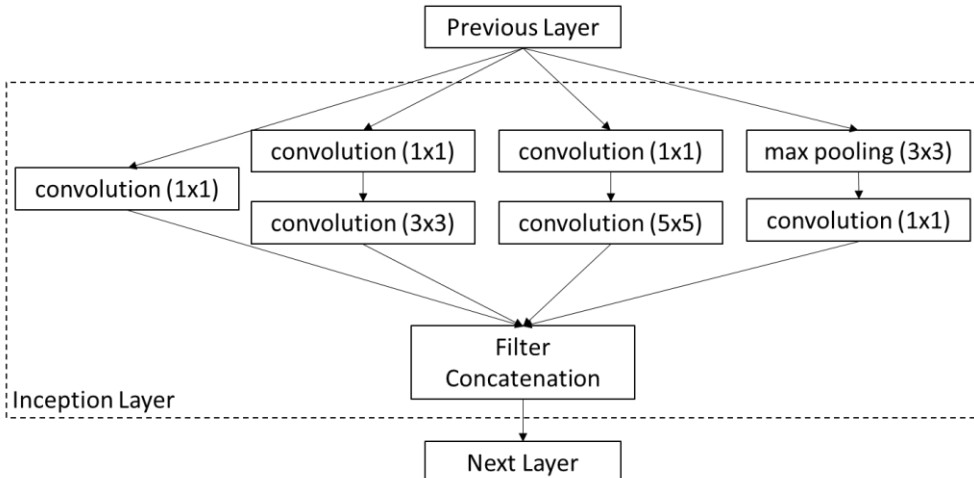

**Figure 7.** Inception layer with dimensionality reduction (adapted from [54]).

Another fundamental difference in GoogLeNet compared to other CNN architectures is its use of sparsity as opposed to fully connected layers. This is based on the foundation introduced by Arora et al. [55] that, "if the probability distribution of the dataset is representable by a large, very sparse deep neural network, then the optimal network topology can be constructed layer after layer by analysing the correlations statistics of the predicting layer activation and clustering neurons with highly correlated outputs" [53]. In GoogLeNet architecture, sparsity is introduced to address the challenges with the computational costs and overfitting associated with fully connected layers.

In this study, GoogLeNet architecture is used as a basis of the 2D CNN algorithm used for the proposed damage detection system. For the purpose of this work, the main hyperparameters of the network are optimised using Bayesian Optimisation to adopt this network for drive-by damage detection purposes. Hyperparameters refer to parameters of the network that are not trainable and are set prior to the training process.

*2.3. Bayesian Optimisation*

The aim of optimising the hyperparameters of the CNN algorithm is to fine-tune the parameters that can return the best performance measured by testing the dataset. The main challenge with hyperparameter optimisation is the high computational cost of the objective function. In each interaction of the hyperparameter search, the CNN needs to be trained and tested. For this type of highly nonlinear problem, a typical grid search and random search can be inefficient and computationally expensive. An efficient alternative to these search methods is the Bayesian approach, which learns from past evaluation results and builds a probabilistic model for the objective function. This approach is able to find global extrema with a considerably small number of objective functions. Given its high performance in addressing optimisation of highly nonlinear nonconvex problems, the method is used in this study to optimise hyperparameters of the model. The alternatives to Bayesian Optimisation are considered to be genetic algorithms and simulated annealing, which are predominantly designed for objectives that are relatively inexpensive to compute.

The Bayesian Optimisation postulates a GP prior, $f(\theta)$ over a latent function, using the mean of zero and a (covariance) kernel matrix expressed as Equation (4):

$$C = \begin{bmatrix} c(\boldsymbol{\theta}_1, \boldsymbol{\theta}_1) & \cdots & c(\boldsymbol{\theta}_1, \boldsymbol{\theta}_i) \\ \vdots & \ddots & \vdots \\ c(\boldsymbol{\theta}_n, \boldsymbol{\theta}_1) & \cdots & c(\boldsymbol{\theta}_n, \boldsymbol{\theta}_n) \end{bmatrix} + \sigma_{noise}^2 \boldsymbol{I} \tag{4}$$

where $c\left(\boldsymbol{\theta}_k, \boldsymbol{\theta}_j\right)$ represents the covariance function and $\sigma_{noise}^2$ standard deviation of Gaussian noise. Assuming $n$ observations of $\mathcal{D}_n = \{(\boldsymbol{\theta}_i, y_i)\}_{i=1}^{n}$, where $\boldsymbol{\theta}_i \in \boldsymbol{\vartheta}$, $y_i = f(\boldsymbol{\theta}_i) + \epsilon_i$,

$\mathbf{Y} = \{y_i\}_{i=1}^n$, $\mathbf{\Theta} = \{\boldsymbol{\theta}_i\}_{i=1}^n$, $\boldsymbol{\theta} = \{\theta_1, \ldots, \theta_d\}$, $d$ is dimension of the hyperparameter vector and $\epsilon_i \sim \mathcal{N}(0, \sigma_{noise}^2)$, the posterior process of $f(\boldsymbol{\theta}_{n+1})|\mathcal{D}_n$ is a Gaussian Process with a mean expressed as Equation (5):

$$\hat{y}(\boldsymbol{\theta}_{n+1}) = \widetilde{C}C^{-1}\mathbf{Y} \tag{5}$$

and covariance of $s^2(\mathbf{x})$, expressed as:

$$s^2(\boldsymbol{\theta}_{n+1}) = c(\boldsymbol{\theta}_{n+1}, \boldsymbol{\theta}_{n+1}) - \widetilde{C}^T C^{-1} \widetilde{C} \tag{6}$$

where $\widetilde{C} = C(\boldsymbol{\theta}_{n+1}, \mathbf{\Theta})$. This implies that the predictive posterior distribution depends heavily on the covariance function $c(\boldsymbol{\theta}_k, \boldsymbol{\theta}_j)$. For the purpose of this study, the automatic relevance determination Matérn 5/2 kernel as defined by Snoeket al. [56] is used.

The key in Bayesian Optimisation is the acquisition function, which determines the trade-off between exploration (high-uncertainty regions) and exploitation (low-value regions) to define the next point of evaluation [57]. A common acquisition function is known as expected improvement ($EI$), which can be expressed as Equation (7):

$$EI(\boldsymbol{\theta}) = \mathbb{E}[I(\boldsymbol{\theta})] = \mathbb{E}[\max(f_{min} - f(\boldsymbol{\theta}), 0)|\{(\boldsymbol{\theta}_i, y_i)\}_{i=1}^n] \tag{7}$$

where $f_{min}$ represents current optimal function value and $I(\boldsymbol{\theta})$ improvement at $\boldsymbol{\theta}$. In Bayesian Optimisation, the optimal point is defined at EI maximum, hence:

$$\hat{\boldsymbol{\theta}} = \underset{\theta \in \boldsymbol{\vartheta}}{\operatorname{argmax}} \, EI(\boldsymbol{\theta}) \tag{8}$$

Further details on Gaussian Optimisation can be found elsewhere [58].

Since the basis of the CNN algorithm used in this study is inherited from the pre-trained GoogLeNet network, the number of hyperparameters is considerably less than a network built anew. In this study, the hyperparameters considered include dropout probability, initial learning rate and the maximum number of epochs. Dropout probability represents the probability of dropping out nodes and corresponding weights in the classification layers. Learning rate determines the change in weight per time, and the maximum number of epochs defines the number of training cycles for each training dataset. For each hyperparameter, a search range is defined to describe the Bayesian Optimisation search domain. In this study, the dropout probability search boundaries are 50–95%, the initial learning rate boundaries are defined from $1 \times 10^{-4}$ to 1 and the epochs can vary from 30 to 80.

Figure 8 shows the optimisation search within the defined domain for all three considered hyperparameters and defined damage scenarios. Each point in this figure represents one search and the intensity of colour for each point shows the value of the objective function, varying from black for zero (0% error in prediction) to white for 1, representing 100% error in prediction. As can be seen from this figure, the combination of an initial learning rate of $5 \times 10^{-4}$, epochs of 31 and dropout probability of 55% result in the maximum number of the optimum objective function values.

Using optimised hyperparameter values, the network was trained and tested. Table 2 summarises the structure of the CNN structure used in this study.

In this study, the mini-batch method is used for training purposes which is, in essence, a combination of stochastic Gradient Descent (updating and adjusting weights immediately after each training round) and batch method (updating weights once the error is calculated for the entire training data). Training time for the selected optimal architecture on a machine with an i9-7940x processor, CPU @3.10 GHz and memory of 32 GB is 24–30 min per 100 scenarios. The following section presents the result of the training process and predictions using the testing dataset.

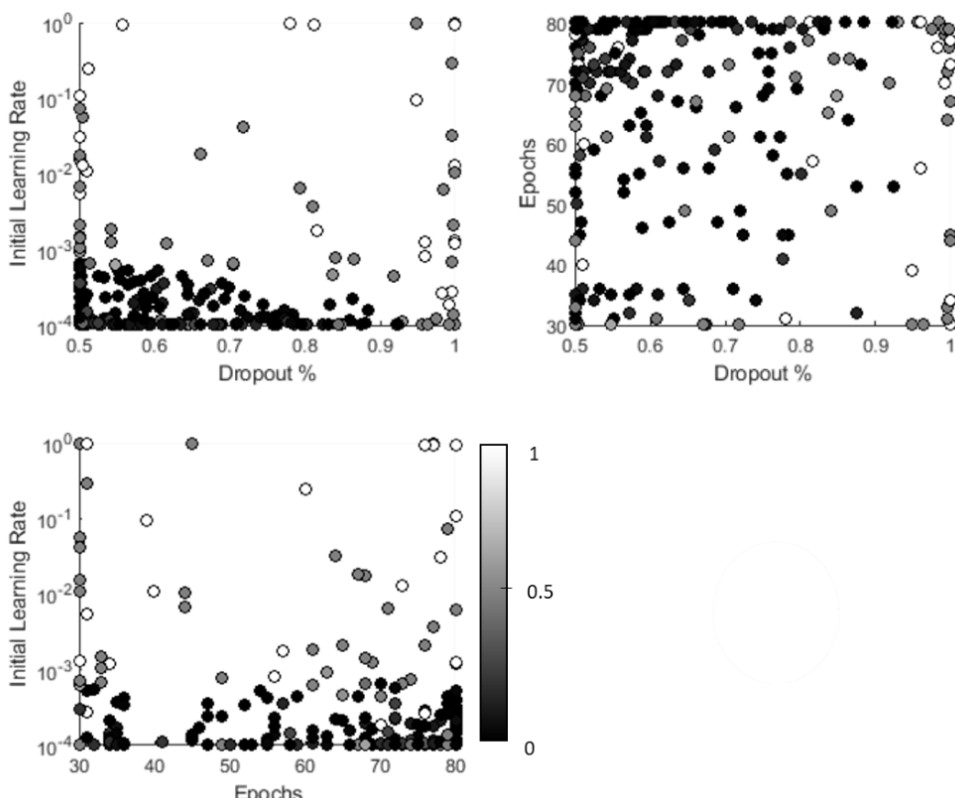

**Figure 8.** Bayesian Optimisation results for hyperparameters of learning rate, maximum number of epochs and dropout probability.

**Table 2.** Adapted GoogLeNet architecture.

| Type |
| --- |
| convolution layer 7 × 7 and stride [2,2] |
| max pool layer 3 × 3 and stride [2,2] |
| convolution layer 3 × 3 and stride [1,1] |
| max pool layer 3 × 3 and stride [2,2] |
| inception (3a) |
| inception (2b) |
| max pool layer 3 × 3 and stride [2,2] |
| inception (4a) |
| inception (4b) |
| inception (4c) |
| inception (4d) |
| inception (4e) |
| max pool layer 3 × 3 and stride [2,2] |
| inception (5a) |
| inception (5b) |
| average pool layer 7 × 7 and stride [1,1] |
| dropout layer with probability of 55% |
| fully connected layer |
| softmax |

## 3. Results

The simulated acceleration and corresponding CWT real-valued coefficients were divided into two sets of training and testing datasets. For this study, 70% of the simulated database is used for training purposes and 30% is held out for testing. Once the algorithm

is trained, the performance of the algorithm is tested using the set of data that has not been seen by the algorithm during the training process. The output of the algorithm is presented in binary classes of healthy and damaged states and the performance is measured based on the accuracy of the predicted state of the bridge.

To better understand how the network decides on bridge healthy state, the gradient-weighted class activation mapping technique (also referred as Grad-CAM localisation mapping) introduced by Selvaraju et al. [59] is used here. In this method, the gradient of classification score with respect to the convolutional features determined by the network is used to highlight the most discriminating parts of input data for classification. The grad-CAM localisation map, $L_{Grad-CAM}^c$, for any class of $c$ can be expressed as Equation (9) [59]:

$$L_{Grad-CAM}^c = f\left(\sum_k \alpha_k^c A^k\right) \tag{9}$$

where $\alpha_k^c$ captures the importance of feature map $k$ for a target class of $c$ and is expressed as Equation (10):

$$\alpha_k^c = \frac{1}{Z}\sum_i\sum_j \frac{\partial y^c}{\partial A_{ij}^k} \tag{10}$$

in which $\frac{\partial y^c}{\partial A_{ij}^k}$ represents gradients of the score for class $c$, $y^c$, with respect to feature maps of a convolutional layer, $A_{ij}^k$. Further details of the approach are provided elsewhere [59]. Figure 9 shows an example of the application of this method to one of the input CWT images for a speed of 105 kph. As shown in Figure 5, the most discriminative part of the image is focused in the region of 5–10 Hz. Figure 9 confirms that the network is correctly focusing on this region as highlighted.

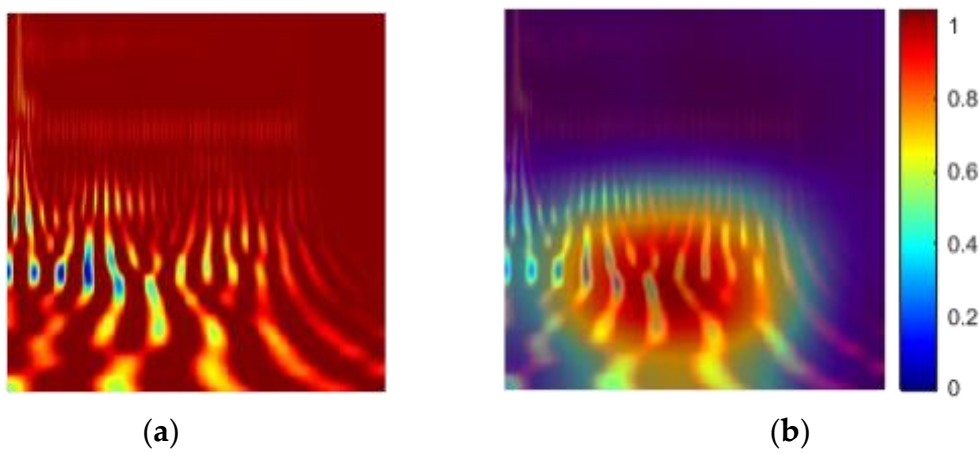

**(a)** **(b)**

**Figure 9.** An example of discriminating cues from CWT images: (**a**) original image; (**b**) highlighted sensitive features.

In a similar attempt to investigate the features that have been most useful in the learning process, Figure 10b,c demonstrate normalised and scaled activation images corresponding to the maximum activating channel for the first pooling layer (i.e., max pool layer 3 × 3) and last inception layer (i.e., inception (5b)), respectively. In Figure 10b,c, each white pixel represents strong positive activation while each black pixel shows negative activation. The first pooling layer is one of the early layers that focuses on low-level features (e.g., edges and colours), while the deeper layers, such as the inception (5a), operate on high-level features such as the difference between the damaged and healthy image.

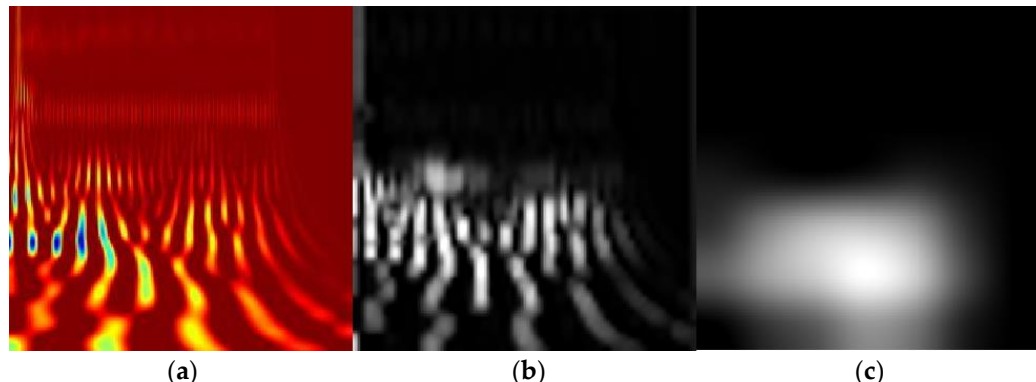

**Figure 10.** An example of discriminating cues from CWT images: (**a**) original image; (**b**) activating features in max pool layer 3 × 3; (**c**) activating features in inception (5b).

To further demonstrate the power of activations in the trained CNN, the t-distributed stochastic neighbour embedding method (t-SNE) [60] is used. This method is often employed to present high-dimensional data in a 2D/3D representation. In simple steps, the t-SNE function generally calculates pairwise distances between high-dimensional points, creates a standard deviation for each point, calculates a similarity matrix with the corresponding joint probability distribution and then creates an initial set of low-dimensional points. This process is iteratively repeated to update the low-dimensional points with the objective of minimizing the Kullback–Leibler divergence between a Gaussian distribution in high-dimensional space and a t-distribution in the low-dimensional space [60].

Figure 11 demonstrates the change in clustering power of the algorithm from the first pooling layer to the final convolution layer and softmax layer using the t-SNE in 2D space. In this representation, the nearby points in 2D space correspond to nearby points in high-dimensional space. Figure 11 shows that while the early layers focus on shallow features, deeper layers detect more complex features by combining features from earlier layers, hence are stronger classifiers. As shown in Figure 4, low-level features in input data do not have the power of clustering (i.e., healthy and damaged images are very similar), which explains the poor performance of early layers.

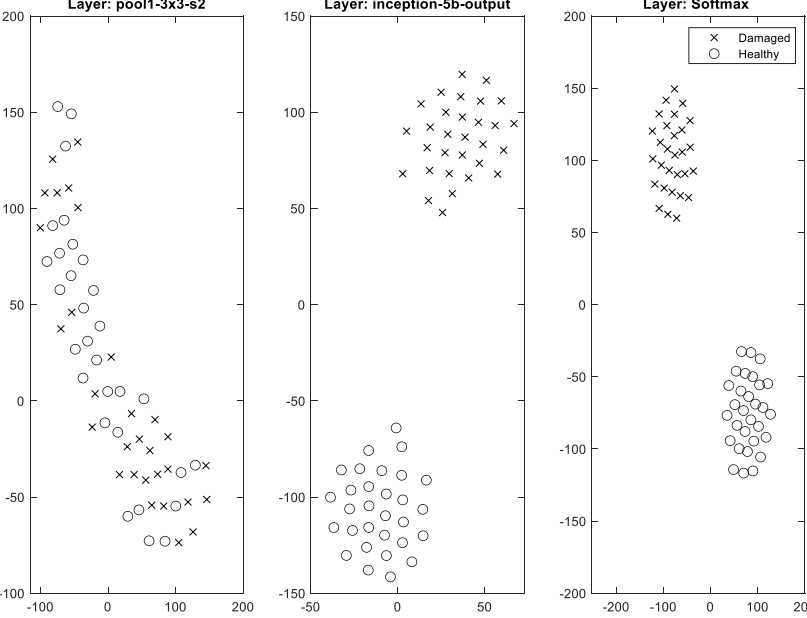

**Figure 11.** An example of network behaviour from first pooling activation layer to final softmax activations.

Figure 12 demonstrates the accuracy of the trained algorithm using the training dataset for all six levels of damage and three damage locations. As it is expected, the accuracy of the algorithm is a function of the severity of the damage (reduction in flexural stiffness), as is shown by the overlaid change in fundamental frequencies for each case. It can be seen that in comparison to damage in quarter-of-span and three-quarter-of-span, damage at mid-span results in a greater change in frequency, which also explains the better performance in the damage detection algorithm for damage scenarios at mid-span. The figure also shows that the algorithm can successfully detect any damage scenario with an impact of more than 2% in natural frequency using train-borne axle acceleration signals while travelling with an average speed of 100 kph.

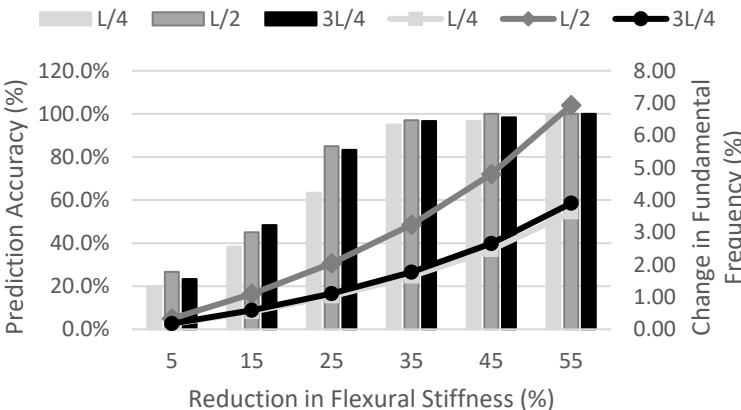

**Figure 12.** Prediction accuracy of damage detection algorithm for different levels of damage and different locations.

To demonstrate the impact of vehicle velocity on the performance of the trained algorithm in detecting damage for simulated scenarios, Figure 13 shows correct (true-positive and true-negative) and incorrect (false-positive and false-negative) predictions as a function of speed. As can be expected, the number of incorrect predictions decreases with the intensity of the damage. It can be seen that the performance of the algorithm is more a function of the number of samples within a certain speed range rather than the speed value. It can be seen that the majority of correct predictions are focused in a speed range of 90–110, which contains 70% of training data, while speeds in the tail range show more frequency of incorrect scenarios. This figure highlights the power of the algorithm even under operational traffic speed, demonstrating the feasibility of the application of the methodology under operational conditions.

*Recommendations for Future Work*

The success of the proposed approach in this study has been investigated under a certain level of variability in operational speed, measurement noise and rail irregularities. However, a detailed examination of the impact of the environmental conditions, such as temperature and humidity, which can contaminate signals and mask the damage-induced signature in the signals, is beyond the scope of this study. Therefore, the impact of varying environmental conditions on the performance of the algorithm requires further investigation.

The current structure of the input data uses raw real-valued CWT coefficients of the first bogie with the assumption that additional supporting information such as vehicle speed and rail irregularities does not exist, to demonstrate the feasibility of the approach in the absence of such information. However, incorporating such information in the architecture of the input structure may improve the accuracy of the algorithm.

The proposed approach demonstrates the feasibility of the algorithm in detecting damage (first level of damage identification). The next step will be to expand the application to higher levels of damage identification, i.e., severity and localisation.

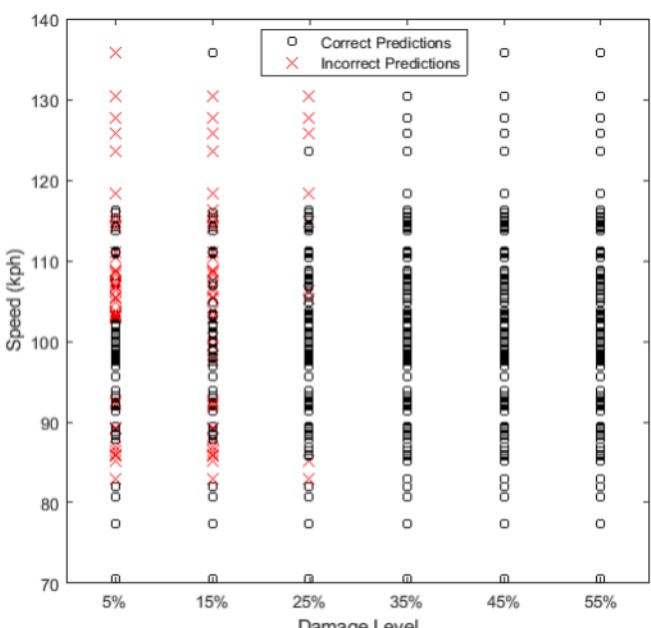

**Figure 13.** True-positive and true-negative predictions for trained algorithms.

## 4. Conclusions

This study presents the application of a 2D CNN structure for drive-by/indirect damage detection. CNN algorithms are well-known for their application in image/object recognition purposes, and in recent years, their application has been extended to vision-based structural health monitoring. This paper presents the first attempt at employing 2D CNN algorithms for vibration-based damage detection using train-borne acceleration signals.

A numerical train–track–bridge interaction model was built and utilised to simulate train accelerations for a range of damage/healthy scenarios under different train speeds and track irregularities. The labelled simulated acceleration signals were then used as raw input. In this study, the well-known pre-trained GoogLeNet architecture was utilised as the basis of the CNN algorithm. The hyperparameters of the algorithm were then fine-tuned for drive-by damage detection purposes using Bayesian Optimisation to ensure model robustness. The performance of the trained algorithm was tested on six different damage intensities at three different locations. The results of the study show that the trained algorithm can successfully predict damage with the impact of more than 2% change in the fundamental natural frequency for all three considered locations.

The power of the proposed approach is in its capacity to detect damage using train-borne signals without the need for direct measurements from the bridge and/or bridge-specific information. Furthermore, the study demonstrates the feasibility of drive-by damage detection under operational speed, utilising shorter bursts of data.

**Funding:** This research received no external funding.

**Informed Consent Statement:** Not applicable.

**Data Availability Statement:** Some or all data, models or code that support the findings of this study are available from the corresponding author upon reasonable request.

**Conflicts of Interest:** The author declares no conflict of interest.

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
