# Peer review of "Deep-Learning-Based Drive-by Damage Detection System for Railway Bridges"

_infrastructures, doi:10.3390/infrastructures7060084_

Round 1

Reviewer 1 Report

Please change the citation format of papers with multiple authors, e.g. "Yang and co-authors [4]" to "Yang et al. [4]".

Author Response

The author thanks the reviewer for their time reviewing the manuscript and providing constructive feedback and has revised the manuscript to address the issue raised.

Reviewer 2 Report

The authors proposed a vibration-based & deep learning-based railway bridge health assessment. The manuscript is adequately written and fairly presented. The following suggestions must be considered to improve the readability of the article.

  1. The first glance through the manuscript prersented a basic flaw in the article. The authors have used black and white figures (which is great), however, not througout. It is suggested that authors change Fig. 9,10,11, & 13 to B&W for consistent formatting.
  2. Why Baysian optimization is used? Why not Adam optimizer? Explaiantion is required.
  3. Why specificaly CNN is used and not LSTM while LSTM is more robust for time-series (acceleration response)?
  4. The authors have used CWT coffecients, it would great if a reason is provided for it, and why not other more robust coefficients such as SynchoSqueezing Tranform are used?
  5. It would be better if Fig. 8 is pesented in a tabular manner. 
  6. The authors should cite the following articles to further improve the readability of the article.

(a) Nex, F., Duarte, D., Tonolo, F. G., and Kerle, N. (2019). Structural Building Damage Detection with Deep Learning: Assessment of a State-of-the-Art CNN in Operational Conditions. Remote Sensing, 11(23), 2765.

(b) Sony, S. 2021. “Towards multiclass damage detection and localization
using limited vibration measurements.” Ph.D. thesis, Department of Civil
and Environmental Engineering, University of Western Ontario, Canada.

(c) Lin, Z., Liu, Y., & Zhou, L. . Damage Detection in a Benchmark Structure Using Long Short-term Memory Networks. 2019 Chinese Automation Congress (CAC). IEEE.

Author Response

The author sincerely thanks the reviewer for their positive and constructive comments and feedback and has revised the manuscript to address the issues raised.

Round 2

Reviewer 2 Report

The authors have clarified and improved the manuscript. I recommend it for publication.